# Fairness Beyond Prediction: Rethinking Alignment Procedures for Agentic AI Systems

## Abstract

As AI systems transition from static predictive models to interactive and agentic systems that plan, adapt, and act over time, classical algorithmic fairness frameworks become insufficient. Traditional fairness notions were largely developed for single-shot prediction or prediction-based decision making, and they often fail to capture procedural, temporal, and emergent fairness issues introduced by alignment procedures such as preference optimization, RLHF, and policy constraints. This paper argues that fairness must be analyzed as a property of *alignment processes* and *agentic dynamics*, not solely model outputs. We identify key fairness failure modes that can arise across value learning, policy adaptation, and long-horizon deployment, including representational asymmetries in preference data, feedback-driven amplification, and path-dependent disparities. We then propose a lightweight conceptual framework for fairness across alignment procedures, outlining evaluation questions and governance implications for building fairer agentic AI systems.

## 1 Introduction

Algorithmic fairness research has historically focused on predictive settings—classification, regression, and ranking—where a model produces an output given an input, and fairness is evaluated via parity constraints across demographic groups (e.g., equalized odds, demographic parity). These approaches assume that decisions are largely *single-shot* and that the system does not substantially adapt its behavior based on interaction.

This assumption increasingly breaks in modern deployments. Contemporary AI systems are often *agentic*: they reason, plan, interact with users and tools, and update their behavior over time. Examples include conversational assistants, tool-augmented agents, personalized tutoring systems, and autonomous decision-support systems. These systems are shaped by *alignment procedures*—such as preference learning, RLHF, instruction tuning, safety fine-tuning, and constraint-based policies—that define how an agent learns what to do and how it adapts in response to feedback.

In this regime, fairness failures rarely arise from a single prediction. Instead, unfairness can emerge *procedurally* through how objectives are learned, how feedback is incorporated, and how adaptation changes downstream behavior. A system may appear fair at one evaluation snapshot yet become unfair after extended interaction, feedback loops, or distributional shifts. As a result, fairness becomes inseparable from alignment: alignment procedures and agentic dynamics determine whose interests are represented, whose preferences shape behavior, and how harms accumulate over time.

This paper asks:

> *How should fairness principles and tools evolve when AI systems not only predict,*
> *but also adapt and act?*

We argue that fairness must be evaluated across the *alignment pipeline itself*. We outline fairness risks introduced by alignment procedures, propose a lightweight framework for fairness across alignment and agentic systems, and provide practical evaluation and governance implications. Our goal is

not to introduce a new metric, but to clarify where existing fairness tools break down and to propose concrete questions to support fairness-aware alignment.

## 2 WHY CLASSICAL FAIRNESS BREAKS FOR AGENTIC SYSTEMS

### 2.1 FROM OUTCOME FAIRNESS TO PROCEDURAL FAIRNESS

Classical fairness notions focus on outcomes (e.g., prediction parity across groups). In agentic systems, the relevant object is often a *policy* that selects actions, observes consequences, and updates based on feedback. This shift introduces three challenges.

**Temporal coupling.** Actions affect future states and opportunities. Even if an agent satisfies a parity constraint at time $t$, its actions may change the environment and create disparities at later times.

**Feedback loops.** Agent behavior influences the data used for continued alignment. Unequal feedback quality or feedback availability across groups can amplify disparities.

**Policy adaptation.** Many agents continuously update. Fairness properties are not static: a fairness guarantee at deployment does not imply fairness after adaptation.

These characteristics suggest that fairness must include *procedural* questions: how the agent learns values, how it updates, and how harms accumulate.

### 2.2 FAIRNESS RISKS INTRODUCED BY ALIGNMENT PROCEDURES

Alignment procedures are often treated as neutral mechanisms for shaping behavior. We argue they are primary sites of fairness risk:

**Preference learning.** Preferences are incomplete, context-dependent, and socially biased. If preference data under-represents marginalized perspectives, the learned objective can encode asymmetrical value weighting even when downstream predictions appear balanced.

**RLHF and reward optimization.** Reward models and feedback providers can contain implicit bias. Optimization can privilege short-term performance signals that correlate with dominant-group norms, leading to systematic disadvantage over repeated interactions.

**Constraint-based alignment.** Constitutional rules or safety constraints can freeze normative assumptions that do not generalize across cultures or contexts. These constraints may be protective for some groups while burdening others.

These failure modes are not fully captured by post-hoc fairness checks on outputs. They require analyzing fairness *within* the alignment pipeline.

## 3 A FRAMEWORK FOR FAIRNESS ACROSS ALIGNMENT PROCEDURES

We propose viewing fairness as a process-level property spanning three stages. This framing supports practical audits and clarifies where interventions should occur.

### 3.1 STAGE I: VALUE LEARNING FAIRNESS

This stage concerns *what* the system is optimized for. Fairness questions include: (i) whose preferences or values are represented, (ii) how conflicting preferences are aggregated, (iii) whether minority or long-tail preferences are preserved, and (iv) whether value learning is robust to strategic or unequal feedback.

A failure at this stage yields a misaligned objective that persists even if downstream behavior is later constrained.

## 3.2 Stage II: Policy Adaptation Fairness

Agentic systems adapt through interaction. Fairness risks include: (i) unequal exploration across user groups, (ii) differential learning rates that favor groups producing more feedback, (iii) path dependence where early interactions lock in biased behaviors, and (iv) shifting behavior under distribution drift.

This motivates fairness evaluation across *learning trajectories*, not only static snapshots.

## 3.3 Stage III: Long-horizon Impact Fairness

Agentic decisions accumulate over time. Small asymmetries can compound into disparities in access, opportunity, or exposure to harm. Long-horizon fairness requires evaluating: (i) cumulative reward or benefit distribution, (ii) compounding error dynamics, (iii) differential risk exposure, and (iv) the downstream societal impacts of repeated agent actions.

This stage connects technical evaluation to governance: fairness becomes an ongoing responsibility, not a one-time certification.

## 4 Implications for Evaluation and Governance

Our framework suggests that fairness evaluation for aligned, agentic systems must extend beyond output metrics.

**Alignment-aware fairness audits.** Audits should examine preference data composition, feedback mechanisms, and reward model behavior. Questions include: which groups contribute feedback, what is optimized, and how updates change behavior across groups.

**Process transparency.** Understanding alignment choices and update rules becomes central to fairness accountability. This includes documenting data sources, preference aggregation methods, and constraints used in alignment.

**Governance upstream.** Oversight and risk management should target alignment procedures (how systems are trained and updated), not only deployed outputs. This is especially important for long-lived agents deployed in sensitive domains.

We emphasize that these recommendations are compatible with existing fairness tools; the key shift is to apply them at *procedural points* in the alignment pipeline and to measure fairness over time.

## 5 Conclusion

As AI systems become increasingly agentic, fairness can no longer be treated as a static property of predictions. It must be understood as a dynamic property of alignment procedures and agentic adaptation. This paper reframes fairness as a procedural concern that spans value learning, policy adaptation, and long-horizon deployment. By identifying where classical fairness breaks down and proposing a process-level framework, we aim to support the development of fairer, more accountable aligned and agentic AI systems.

## References

## A LLM Usage Disclosure

Large language models were used for language refinement and formatting assistance. All conceptual framing, arguments, and technical content were authored and verified by the human authors.

