# OpenReview forum: "Fairness Beyond Prediction: Rethinking Alignment Procedures for Agentic AI Systems"
_ICLR.cc/2026/Workshop/AFAA — Submitted to AFAA 2026_

### Official Review · Reviewer_wWpv · 2026-02-10
**Hard to gauge the soundness of the work**

**Rating:** 2
**Confidence:** 4

**Summary:**

The paper questions the current whether the approach of outcome fairness should be applied when evaluating agentic systems. They argue that a one-time evaluation as is done for most fairness evaluation approaches is insufficient in the changing environment of an agentic system. Rather a framework should be applied that evaluates the fairness of a system at different points in its lifecycle.

**Strengths:**

The paper identifies some key elements for which traditional ML applications differ from the agentic setting.
It further showed some interesting key points on when agentic system should be evaluated and some of the questions that could be asked for specific points in the lifecycle.

**Weaknesses:**

The weaknesses of the paper are likely due to the short nature of the tiny paper, however the information provided in the tiny paper is insufficient for me to gauge the soundness of the work.

The tiny paper has no citations, even though the citations did not count toward the page count. While a full section on related work is not required some indication of positioning this work within the broader research field would have been appropriate. Coupled to that, the authors make several claims throughout the paper such as "Unequal feedback quality or feedback availability across groups can amplify disparities.", which intuitively would seem correct, but these statements are best supported by research highlighting this behavior.

The paper says that it proposes a framework/audit for evaluation. However, only some example questions are provided for each stage in the lifecycle. While the concept should not be worked out completely, an approach or at least idea of what the full framework would look like would have made the contribution clearer. Is this framework an automatic framework testing things like "whether value learning is robust to strategic or unequal feedback" or is it more a documentation scheme where the creators of an agentic system should document their answers to questions such as "whose preferences or values are represented".

---

### Official Review · Reviewer_SYg7 · 2026-02-20
**From Static to Dynamic Evaluation**

**Rating:** 2
**Confidence:** 3

**Summary:**

The short paper discusses how evaluation / alignment needs to be treated differently for agentic and continuously updating systems as opposed to "classic" prediction settings. In particular, the paper poses the question of how principles and tools of algorithmic need to evolve / adapt for AI systems which are able to adapt and act (beyond just prediction). They argue that in this context, fairness needs to be evaluated across the alignment pipeline ifself. After briefly describing failure modes of existing approaches, the paper introduces a small framework comprised of three stages.

**Strengths:**

- The central question of the short paper, how fairness should evolve for agentic settings beyond prediction, is interesting and worthy of examination.
- Relatedly, using the alignment *pipeline* as a lense of examination seems like a worthwhile idea and the central conclusion of the paper, that the fairness of agentic AI systems should be seen as a dynamic property makes sense.

**Weaknesses:**

- The short paper does not provide any references. As references do not count towards the page limit, I do not see any compelling reason as to why the authors made this choice. Referencing prior work is foundational to the academic process and I believe this work would benefit from stronger engagement with the literature.
	- This is the only short paper I am reviewing, so I double checked the CfP to confirm that references in short papers are allowed (if not, I would welcome a correction by one of the chairs or the authors and would update my review accordingly).
- Relatedly many of the claims in the short paper are wide ranging and not supported. Oftentimes single examples are used to argue its case without addressing alternatives at all (e.g. L063 onwards, L079 onwards, ...).
- The paper is quite vague and does not concretely define the scope of its framework. Does this apply to any agentic system, to any decision making system, to any decision making system with agentic properties? Terminology is not clearly defined, nor are concrete examples provided. What is the understanding of an agentic system in this case? What does fairness of such a system mean?
- While multiple examples and points are provided it seems the central conclusion of the paper is that continuously running and potentially self-updating systems need to also be continuously evaluated. This conclusion, however, has already been proposed and discussed in large bodies of prior work.

---

### Meta-Review · Area_Chair_7h94 · 2026-02-24

**Recommendation:** Reject
**Confidence:** 5

**Metareview:**

This paper aims to address the questions of how fairness principles and tools should evolve for adaptive and agentic AI systems as opposed to static predictive systems. This is indeed a timely and important topic. However, I concur with the reviewers that, for a workshop submission, the paper lacks sufficient detail to be impactful for readers and discussion at the workshop. For instance, the paper includes no citations. I encourage the authors to continue working on this important topic. A developed version of this work could be impactful.

---

### Decision · Program_Chairs · 2026-03-02

Reject